Anhanguera taxonomy revisited: is our understanding of Santana Group pterosaur diversity biased by poor biological and stratigraphic control?

Pinheiro Felipe L. felipepinheiro@unipampa.edu.br 1
Rodrigues Taissa 2
1 Laboratório de Paleobiologia, Universidade Federal do Pampa , São Gabriel , RS , Brazil
2 Departamento de Ciências Biológicas, Universidade Federal do Espírito Santo , Vitória , ES , Brazil
Piñeiro Graciela
Electronic publication date: 2017 May 4
Publication date: 2017
Volume: 5
Electronic Location ID: e3285
Received 2017 Jan 4; Accepted 2017 Apr 8
Copyright: ©2017 Pinheiro and Rodrigues
Copyright year: 2017
Copyright holder: Pinheiro and Rodrigues
License: This is an open access article distributed under the terms of the Creative Commons Attribution License, which permits unrestricted use, distribution, reproduction and adaptation in any medium and for any purpose provided that it is properly attributed. For attribution, the original author(s), title, publication source (PeerJ) and either DOI or URL of the article must be cited.
License URL: https://creativecommons.org/licenses/by/4.0/

Keywords: Pterosauria, Anhangueridae, Romualdo formation, Cretaceous, Allometry, Geometric morphometrics

Funding: Deutscher Akademischer Austausch Dienst (DAAD) CNPq 140407/2007–3 290019/2008–7 460784/2014-5 DAAD A/0871633 FAPES 67678254/15 Richard Gilder Graduate School Collection Study SYNTHESYS Project NL–TAF–91 Study of specimen AMNH 22555 was funded by a Collection Study Grant (Richard Gilder Graduate School), whereas research at the BSP and other European collections was supported by a Deutscher Akademischer Austausch Dienst (DAAD) scholarship granted to FLP. TR received funding from CNPq (grants 140407/2007–3, 290019/2008–7, and 460784/2014-5), DAAD (grant A/08 71633), FAPES (grant 67678254/15), Richard Gilder Graduate School Collection Study Grant, and SYNTHESYS Project (grant NL–TAF–91). The funders had no role in study design, data collection and analysis, decision to publish, or preparation of the manuscript.

==============================
Background

Anhanguerids comprise an important clade of pterosaurs, mostly known from dozens of three-dimensionally preserved specimens recovered from the Lower Cretaceous Romualdo Formation (northeastern Brazil). They are remarkably diverse in this sedimentary unit, with eight named species, six of them belonging to the genus Anhanguera. However, such diversity is likely overestimated, as these species have been historically diagnosed based on subtle differences, mainly based on the shape and position of the cranial crest. In spite of that, recently discovered pterosaur taxa represented by large numbers of individuals, including juveniles and adults, as well as presumed males and females, have crests of sizes and shapes that are either ontogenetically variable or sexually dimorphic.

Methods

We describe in detail the skull of one of the most complete specimens referred to Anhanguera, AMNH 22555, and use it as a case study to review the diversity of anhanguerids from the Romualdo Formation. In order to accomplish that, a geometric morphometric analysis was performed to assess size-dependent characters with respect to the premaxillary crest in the 12 most complete skulls bearing crests that are referred in, or related to, this clade, almost all of them analyzed first hand.

Results

Geometric morphometric regression of shape on centroid size was highly statistically significant (p = 0.0091) and showed that allometry accounts for 25.7% of total shape variation between skulls of different centroid sizes. Premaxillary crests are both taller and anteroposteriorly longer in larger skulls, a feature consistent with ontogenetic growth. A new diagnosis is proposed for Anhanguera, including traits that are nowadays known to be widespread within the genus, as well as ontogenetic changes. AMNH 22555 cannot be referred to “Anhanguera santanae” and, in fact, “Anhanguera santanae”, “Anhanguera araripensis”, and “Anhanguera robustus” are here considered nomina dubia.

Discussion

Historically, minor differences in crest morphology have been used in the definition of new anhanguerid species. Nowadays, this practice resulted in a considerable difficulty in referring well-preserved skulls into known taxa. When several specimens are analyzed, morphologies previously believed to be disparate are, in fact, separated by a continuum, and are thus better explained as individual or temporal variations. Stratigraphically controlled excavations on the Romualdo Formation have showed evidence for faunal turnover regarding fish communities. It is thus possible that some of the pterosaurs from this unit were not coeval, and might even represent anagenetic morphotypes. Unfortunately, amateur collecting of Romualdo Formation fossils, aimed especially at commerce, resulted in the lack of stratigraphic data of virtually all its pterosaurs and precludes testing of these further hypotheses.

Introduction

Anhangueridae is a clade of pterosaurs currently known from multiple localities worldwide, including named species from Brazil, the United States, Morocco, China and England (Rodrigues & Kellner, 2013). The majority of identifiable material comes from the Romualdo Formation (Araripe Basin, northeastern Brazil), a well-known fossil Lagerstätte where they are the most abundant and speciose clade of tetrapods, with eight named species (Tropeognathus mesembrinus, Maaradactylus kellneri and six species of Anhanguera) (Table 1), as well as several closely-related pterosaur taxa and dozens of referred specimens. Even though this anhanguerid taxonomy has already been disputed by several authors (Kellner & Tomida, 2000; Fastnacht, 2001; Unwin, 2001; Veldmeijer, 2003), the apparent species diversity seems nonetheless remarkable.

Although the first descriptions of pterosaurs from the Romualdo Formation date from as early as the 1970s (Price, 1971), well-preserved skull material only began to be described in the 1980s and 1990s. In February 1985, Wellnhofer described a number of specimens from the Romualdo Formation, naming two new species based on fossils comprising skull material: “Santanadactylus” araripensis and “Araripesaurus” santanae; both genera were previously described based only on postcranial material. Later that same year, Campos and Kellner described the new genus and species Anhanguera blittersdorffi, based on a complete skull. In 1987, Wellnhofer described two further species, Tropeognathus mesembrinus and “Tropeognathus robustus”. With increasing knowledge of these Romualdo Formation anhanguerids, some new taxonomic proposals arose, including placing all of these species in the genus Anhanguera (Kellner, 1990).

Additional anhanguerid specimens, but no newly named species, were subsequently described by Wellnhofer (1991); among them AMNH 22555 is an incomplete skeleton including a skull and a fragmentary mandible. It was the most complete skeleton then known from the Romualdo Formation, and served as the basis for the first anhanguerid skeleton reconstruction ever made (Wellnhofer, 1991). This specimen was regarded by Wellnhofer (1991) as conspecific with the holotype of “Anhanguera santanae” (previously in the genus “Araripesaurus”). Remarkably, two other almost complete skeletons, including skulls, were later described and referred to the species Anhanguera piscator (Kellner & Tomida, 2000) and “Coloborhynchus” spielbergi (Veldmeijer, 2003).

Table 1 Anhanguera taxonomy.

Synopsis of the named species of Anhanguera from the Romualdo Formation.

Species	Taxonomic status in the present work	Known specimens	Diagnosis	
Anhanguera blittersdorffiCampos & Kellner, 1985	Type-species	MN 4805-V (holotype) Pz-DBAV UERJ 40	Large number (52) of alveoli on the upper jaw	
Anhanguera araripensis (Wellnhofer, 1985)	nomen dubium	SNSB-BSPG 1982 I 89 (holotype)	Non diagnostic	
Anhanguera santanae (Wellnhofer, 1985)	nomen dubium	SNSB-BSPG 1982 I 90 (holotype)	Non diagnostic	
Anhanguera robustus (Wellnhofer, 1987)	nomen dubium	SNSB-BSPG 1987 I 47 (holotype)	Non diagnostic	
Anhanguera piscatorKellner & Tomida, 2000	Valid	NSM-PV 19892 (holotype)	From Kellner & Tomida (2000): middle part of the basisphenoid presents a constriction; neural spine of the axis forms a 45°angle; distal articulation of the ulna bears a sharp ventral crest; shaft of the scapula is constricted; coracoid has a small cranial process; caudal vertebrae are elongate; neural spines of the middle caudal vertebrae reach the preceding vertebra; neural spines of the middle caudal vertebrae have a well developed ventral process.	
Anhanguera spielbergi (Veldmeijer, 2003)	Valid	RGM 401 880 (holotype)	From Veldmeijer (2003): mandibular groove does not extend to the distal lateral expansion of the mandible; sternal plate triangular and as long as wide.	

Table 2 Other specimens of Anhanguera known by complete or almost complete skulls, and which have been described or cited in the literature.

Specimen	Taxonomic history	Taxonomic status in the present work	
AMNH 22555	Referred to “Anhanguera santanae” by Wellnhofer (1991)	Anhanguera sp.	
MN 4735-V	Referred to “Anhanguera araripensis” by Kellner & Tomida (2000)	Anhanguera sp.	
NHMUK R 11978	None	Anhanguera sp.	
SAO 16494	Referred to “Coloborhynchus araripensis” by Veldmeijer, Meijer & Signore (2006)	Anhanguera sp.	
SMNK PAL 1136	Referred to Anhanguera by Frey & Martil (1994)	Anhanguera sp.	

Today, several skulls (both described and undescribed) are hosted in a myriad of publically accessible collections and thus enabling the examination of a larger sample of Romualdo anhanguerids (Tables 1 and 2). Recent proposals (Kellner & Tomida, 2000; Rodrigues & Kellner, 2008) referred the species A. blittersdorffi, “A. araripensis”, “A. santanae”, “A. robustus”, A. piscator and A. spielbergi to the genus Anhanguera. These taxa are mostly diagnosed by subtle differences in cranial anatomy, mainly focused on the morphology and position of the cranial crest, a character that is presumably sexually dimorphic and/or ontogenically variable (Bennett, 1992; Manzig et al., 2014; Wang et al., 2014). The supposedly diagnostic features of individual Anhanguera species are so discreet and ambiguous that it is virtually impossible to attribute new material to any of the proposed existing taxa with any level of certainty, which also indicates a probable artificial inflation of the diversity of species within the genus. This issue is due to our relatively poor understanding of intraspecific variation in Anhanguera, and which characters might vary according to differences in sex and ontogeny.

Here we reanalyze the skull of the specimen AMNH 22555, originally referred to “Anhanguera santanae” by Wellnhofer (1991), an assumption that was thereafter echoed by other authors (e.g., Kellner & Tomida, 2000; Veldmeijer, 2003). A new description is justified by the fact that Wellnhofer (1991), assuming that AMNH 22555 was not significantly different from “A. santanae” holotype, only devoted one paragraph for the skull in its original description. The new description of AMNH 22555 presented here is used as the basis to explore possible reasons behind the problematic taxonomy of Anhanguera. We apply a geometric morphometric approach to establish size-dependent characters within Anhanguera-like pterosaurs, and make a reassessment of the putative diagnostic features of each of the proposed Anhanguera species, resulting in a revised taxonomy for the genus. We also discuss the possibility that our poor understanding of Romualdo stratigraphy is undermining our wider knowledge of Santana Group pterosaur diversity, by occluding a putative connection between different Anhanguera morphotypes and temporally distinct fossil-bearing strata.

Materials and Methods

Geological setting

All the specimens up until now assigned to the genus Anhanguera have come from the Romualdo Formation (Albian) of the Araripe Basin, northeastern Brazil (Fig. 1). The Romualdo Formation is characterized by conglomeratic sandstones overlain by a transgressive sequence of green and black shales (Assine, 2007). Within the black shales, the presence of several layers rich in carbonate concretions is apparent, and with lateral continuity throughout the basin (Fara et al., 2005; Saraiva et al., 2007; Vila Nova et al., 2011). The genesis of these layers is associated with mass mortality events, followed by the formation of early diagenetic concretions that entrapped a large number of elements of the Romualdo biota.

Figure 1 Location map of the Araripe Basin, northeastern Brazil and simplified stratigraphic chart of the Santana Group.

Levels where pterosaur fossils are found are indicated. Modified from Pinheiro & Schultz (2012).

Studied material

In order to assess the biological and stratigraphic biases that may have impacted on the taxonomy of Anhanguera, we reevaluated the specimen AMNH 22555 (commonly referred as “Anhanguera santanae”, Fig. 2) through a comprehensive cranial description. Although this particular specimen has often been mentioned and illustrated in specialized literature (e.g., Wellnhofer, 1991; Kellner & Tomida, 2000), a detailed description is still pending and, as will be demonstrated, its attribution to “Anhanguera santanae” is mainly based on a superficial resemblance. AMNH 22555 is a partial pterosaur skeleton, composed of an almost complete skull, proximal end of the right mandibular ramus (Figs. 2F and 2G), nearly all vertebral elements (Figs. 2A–2E), some ribs, scapulae and coracoids (Figs. 2H and 2I), an almost complete pelvis and some limb elements, including carpals (Figs. 2J and 2K), metacarpals, femoral and humeral fragments, incomplete radius and ulna, pteroid, and foot phalanges (Fig. 2). With the sole exception of Anhanguera piscator (which was accessed through the cast MN 5023-V) and Maaradactylus kellneri (holotype MPSC R 2357), all other specimens here used for comparison and allometric regressions were examined first hand by the authors.

Figure 2 Specimen AMNH 22555, a partial anhanguerid skeleton. Some selected elements are figured in detail.

(A) pelvic region in dorsal view; (B) torso in dorsal view; (C, D, E) sixth cervical vertebrae in, respectively, anterior, dorsal and right lateral views; (F, G) right mandibular ramus in, respectively, medial and lateral views; (H) left scapula in dorsal view; (I) left coracoid in lateral view; (J) distal carpals in distal view; (K) proximal carpals in distal view. Scale bars equal to 50 mm. Line drawings of some bones were modified from Witton (2013).

Allometric regressions

In order to assess size-dependent characters within Anhanguera-like pterodactyloids, we used geometric morphometrics in a series of 12 skulls attributed to Anhanguera and closely-related taxa (Anhangueria sensu Rodrigues & Kellner, 2013), namely: Anhanguera blittersdorffi (holotype, MN 4805-V), Anhanguera piscator (holotype, NSM-PV 19892), Anhanguera spielbergi (holotype, RGM 401 880), Anhanguera sp. (NHMUK R 11978), Anhanguera sp. (SAO 16494), Anhanguera sp. (SMNK PAL 1136), Anhanguera sp. (MN 4735-V, referred to “Anhanguera araripensis” by Kellner & Tomida, 2000), SMNK PAL 3895 (referred to Cearadactylus atrox by Campos, Headden & Frey 2013), Barbosania gracilirostris (holotype, MHNS/00/85), Maaradactylus kellneri (holotype, MPSC R 2357, based on the reconstruction provided by Bantim et al., 2014), Tropeognathus mesembrinus (holotype, SNSB-BSPG 1987 I 46) and Tropeognathus cf. T. mesembrinus (MN 6594-V, based on the reconstruction provided by Kellner et al., 2013). “Anhanguera santanae” (holotype, SNSB-BSPG 1982 I 90), AMNH 22555 (referred to “Anhanguera santanae” by Wellnhofer 1991), and “Anhanguera araripensis” (holotype, SNSB-BSPG 1982 I 89) were not included because the crest is not preserved in these specimens.

Two-dimensional coordinates were captured for 17 landmarks using digital photographs of specimens in lateral aspect and the software TPSDig (Rohlf, 2010). Landmarks were chosen as follow: 1, posteriormost edge of squamosal; 2, dorsalmost edge of the frontoparietal crest; 3, contact between prefrontal and supraorbital, at the dorsal margin of the orbit; 4, contact between jugal and lacrimal; 5, posterior limit of the lateral shelf of the jugal, at the base of the ascending process of this bone; 6, contact between frontoparietal and postorbital, at the posterior margin of the orbit; 7, ventral edge of the quadrate; 8, anterior limit of the lateral shelf of the jugal, at the base of the ascending process of this bone; 9, contact between lacrimal and nasal, at the dorsal margin of the nasoantorbital fenestra; 10, contact between premaxilla and maxilla, at the anterior margin of the nasoantorbital fenestra; 11, posterior extension of the premaxillary crest; 12, dorsalmost extension of the premaxillary crest; 13, mid-length between landmarks 11 and 12, as projected on the dorsal margin of the premaxillary crest; 14, anterior extension of the premaxillary crest; 15, mid-length between landmarks 12 and 14, as projected on the dorsal margin of the premaxillary crest; 16, anterior tip of the rostrum; 17, mid-length between landmarks 7 and 16, as projected on the ventral margin of the maxilla (Fig. 3).

Figure 3 Geometric morphometric analysis of twelve skulls referable to Anhanguera (red dots) and closely related taxa (blue dots) of the regression score on centroid size log.

Used landmarks are plotted in the skull of Anhanguera blittersdorffi holotype.

The main goal of our analyses was to detect and describe morphologic variation attributable to the increase of skull size, especially with respect to the premaxillary crest. Although our study is mainly focused on the genus Anhanguera, the inclusion of closely-related taxa bearing premaxillary ornaments was justified by the assumption that homologous structures in phylogenetically related animals probably shared functions and growth patterns. All analyses were carried out using the MorphoJ software package, version 1.06a (Klingenberg, 2011). The allometric regression included centroid size as a proxy for cranial size (independent variable) and the shape score s proposed by Drake & Klingenberg (2008) (dependent variable), which includes shape changes predicted by allometry, as well as residual variations that are not dependent to size. The MorphoJ algorithm allowed us to then identify morphological changes entirely related to allometry from the residual variations. A permutation test against the null hypothesis of independence was made in order to test the sensitivity of the regression analyses (10,000 rounds).

As the landmark plotting for Maaradactylus kellneri (MPSC R 2357) and Tropeognathus cf. T. mesembrinus (MN 6594-V) was based on tentative reconstructions provided in the literature, respectively by Bantim et al. (2014) and Kellner et al. (2013), a second regression analysis with the exclusion of those specimens was also performed.

Bantim, Saraiva & Sayão (2015) also carried out allometric regressions in order to investigate cranial crest development within Anhangueridae. These authors, however, used a limited sample of six specimens and restricted their analyses to linear values of crest length and height in order to assess morphology.

Results

Allometric regressions

Our first analysis, including the whole sample of 12 skulls attributed to Anhanguera and closely related taxa, detected a highly statistically significant (p = 0.0091) regression of shape on centroid size. Allometry alone accounts for 25.7% of the total shape variation between skulls of different centroid sizes (Fig. 3). The pattern of allometric growth shows a pronounced dorsal shift of landmarks associated to the premaxillary crest (12, 13 and 15), demonstrating a clear trend of dorsal growth of this structure along with the increase in size. It is also evident that landmarks related to the posterior and anterior limits of the premaxillary crest are, respectively, posteriorly and anteriorly displaced in larger specimens (Fig. 3). This pattern of anteroposterior growth of the crest means that larger specimens of Anhanguera-like pterosaurs tend to have premaxillary crests beginning closer to the nasoantorbital fenestra than smaller ones. The distance between the anterior extension of the nasoantorbital fenestra and the posterior end of the crest is also affected by the occurrence of a proportionally longer nasoantorbital fenestra in larger specimens. Also, the anterior end of the crest presents a positive trend of displacement towards the anterior tip of the rostrum in larger skulls. Notably, the orbits show negative allometric growth, with larger specimens bearing proportionally smaller orbits.

The second analysis, in which Maaradactylus kellneri (MPSC R 2357) and Tropeognathus cf. mesembrinus (MN 6594-V) were excluded, also demonstrates a strong relationship between centroid size and shape, with allometry accounting for 22.73% of total shape variation. This second regression was, however, less statistically significant (p = 0.058), but all of the morphological trends detected in the first analysis were still recovered.

We also analyzed the residual (uncorrelated with size) component of variation for each specimen, in an attempt to identify individual morphological disparity, which is potentially attributable to interspecific variation. At least two specimens indeed show a considerable amount of residual variation of shape, unpredicted by our regression model. Specimen MN 4735-V, attributed by Kellner & Tomida (2000) to “Anhanguera araripensis”, for instance, has a much bigger premaxillary crest than what would be expected for an animal of its size class, while the Anhanguera piscator holotype (NSM-PV 19892) has a proportionally small crest for its size. Notably, some of the residual variation observed in other specimens is attributable to diagenetic modification of fossils, such as an upward shift of the rostrum in NHMUK R 11978 and in the Maaradactylus kellneri holotype (MPSC R 2357). Most of the observed residual components of variation, however, are difficult to describe as discrete traits and seem to vary continuously on our sample, with disparate morphologies linked together by a set of intermediaries.

The skull of AMNH 22555

Pterosauria Kaup, 1834	
Pterodactyloidea Plieninger, 1901	
Anhangueria Rodrigues & Kellner, 2013	
Anhangueridae Campos & Kellner, 1985	
AnhangueraCampos & Kellner, 1985	

Anhanguera sp.

Locality and horizon. Romualdo Formation, Araripe Basin, Albian, northeastern Brazil. According to Wellnhofer (1991), the specimen comes from Jardim municipality in the state of Ceará (previously Barra do Jardim), but its exact locality is undetermined.

Figure 4 Interpretative drawings of AMNH 22555 skull.

(A) right lateral, (B) dorsal and (C) palatal views. Abbreviations: ch, choanae; ec, ectopterygoid; fp, frontoparietal; j, jugal; l, lacrimal; m, maxilla; n, nasal; naof, nasoantorbital fenestra; op, opisthotic; pf, prefrontal; pl, palatine; po, postorbital; pm, premaxilla; pt, pterygoid; q, quadrate; so, supraorbital; sq, squamosal; v, vomers. Scale bar equals 100 mm.

Anatomical description. The skull of AMNH 22555 is nearly complete and best preserved in right lateral aspect (Figs. 4 and 5). Even so, rostral elements anterior to the nasoantorbital fenestrae are crushed and laterally compressed in this view. Posterior skull bones are broken and disarticulated in the left lateral view, in which the absence of bones such as the left jugal and lacrimal obliterates the edges of skull openings. The palate anterior to the choanae is well preserved, whereas posterior palatal bones are mostly absent. Parts of the right pterygoid lie inside the nasoantorbital opening in lateral view. Despite the fact that the alveolar margin of the maxillae is intact anteriorly, with the presence of some in situ teeth (mostly broken) and empty alveoli, the ventral margins of both the left and right maxillae are eroded and incomplete posteriorly, preventing an accurate estimation of the total number of tooth positions. The posterior skull roof is almost intact, with a slight lateral displacement of the frontoparietals. Above the nasoantorbital openings, the outer bone layer of the dorsal margin of the fused premaxillae is eroded. The dorsal limits of the premaxillae are badly crushed throughout the anterior half of the skull, preventing the reconstruction of the sagittal crest anatomy. In occipital view, only the broad supraoccipital plate and right opisthotic are fairly well preserved.

In general, the skull bones are disarticulated and, sometimes, displaced from their original positions. The premaxillae and maxillae, as well as the frontals and parietals, are tightly fused with each other, displaying the ordinary condition for pterodactyloids. Some postcranial bones, known to fuse in mature individuals, show the unfused condition in AMNH 22555, indicating that this specimen is osteologically immature (Wellnhofer, 1991; Bennett, 1993). Those elements include separate scapulae and coracoids, as well as proximal and distal carpals (Figs. 2H–2K). The first five dorsal vertebrae show very thick neural spines and prezygapophyses fused with the postzygapophyses of the adjacent vertebra, indicating that a notarium was present in mature individuals of this species (Fig. 2B).

Figure 5 Comparison between the skulls of AMNH 22555 and “Anhanguera santanae” holotype (SNSB-BSPG 1982 I 90).

(A) AMNH 22555 skull in lateral view; (B) Interpretative drawing of the photo in (A). (C) “Anhanguera santanae” (SNSB-BSPG 1982 I 90) skull in lateral view (mirrored); (D) Interpretative drawing of the photo in (C). (E–H) palatal views and interpretative drawings of, respectively, AMNH 22555 and “A. santanae” (SNSB-BSPG 1982 I 90) skulls; (I, J), interpretative drawings of the occipital views of, respectively, AMNH 22555 and “A. santanae” (SNSB-BSPG 1982 I 90) skulls. Abbreviations: ch, choanae; fpc, frontoparietal crest; lpj, lacrimal process of the jugal; pr, palatal ridge; ptf, posttemporal fenestra; soc, supraoccipital crest. Scale bar equal to 100 mm in (A–H) and 50 mm in (I, J).

Premaxilla. The fused premaxillae comprise most of the skull roof, with their posterior ends dorsal to the orbits, where they contact the frontoparietals posteriorly. Although the left premaxilla is considerably well preserved throughout its whole extension, the right element is badly crushed anteriorly to the nasoantorbital fenestra. Sutures between the premaxillae and maxillae can only be observed close to the nasoantorbital fenestrae, especially on the left side of the skull (where this region is best preserved). Anteriorly, the ventral limits of the premaxillae are not clear, and the number of tooth positions associated with these bones cannot be inferred. The dorsal surface of the premaxillae is broken in the region anterior to the nasoantorbital fenestrae, making it difficult to determine if a sagittal crest was present. However, this broken dorsal border extends above the projection of the surface dorsal to the nasoantorbital openings, which may indicate that the crest was present. It is probable that the premaxillae also composed the anterior part of the palate, where the bone is strongly pierced by small foramina. However, due to bone fusion, it is impossible to determine the exact contribution of the premaxillae to the palatal surface. There is a discrete anterior expansion of the skull, with the rostrum being about 1.5–2 mm wider at the level of the 4th tooth sockets than at the 3rd and 5th alveoli. This is more reminiscent of the slight expansion seen in Tropeognathus mesembrinus, but at this point it cannot be ruled out that the expansion could grow larger with maturity.

Maxilla. Bordered dorsally by the premaxillae, the maxillae form the anterior and part of the ventral margins of the nasoantorbital fenestrae. Because the suture lines between the maxillae and premaxillae are located at the anterodorsal border of the nasoantorbital fenestrae, the maxillae also make a small contribution to the dorsal margin of these openings. Ventrally, the palatal plates of the maxillae fuse together (see Ősi et al., 2010; Pinheiro & Schultz, 2012), forming a well-developed palatal ridge that ends about 50 mm before the anterior limits of the choanae. The dental margins of the maxillae form strong rims, and some of the rostral teeth (especially the 7th to 10th tooth pairs) are surrounded at their bases by robust bony collars, generally punctured by foramina on their lingual side. Because the jugal processes of both maxillae are broken, the posterior limits of these bones cannot be determined. Anterior to the 9th tooth pair, the ventral margins of the maxillae gently curve upwards, and the anteriormost teeth are inserted at level with the ventral margins of the orbits.

Nasal. Together with the lacrimals, the nasals form the posterodorsal margins of the nasoantorbital fenestrae. The right nasal is better preserved than the left one, and shows an irregular shape, with acute anterior and posterior extensions. The dorsal margin is straight and contacts the premaxillae. The nasals have lateral longitudinal ridges, probably indicating the contact area with the lacrimals (in AMNH 22555, these bones are slightly displaced). The nasals have concave posterior margins, fitting the convex prefrontals and supraorbitals. The ventral surfaces of the acute anterior processes of the nasals are perforated by well-developed foramina. The nasoantorbital openings are completely filled with carbonaceous matrix and, thus, the medial contact between the left and right nasals, as well as the ventral nasal process, are obscured.

Prefrontal. Only the right prefrontal is preserved. Dorsally, this bone makes contact with the nasal and the supraorbital, whereas ventrally it shows a rectilinear suture with the lacrimal. The prefrontal contributes to part of the anterodorsal margin of the orbit.

Supraorbital. Both supraorbitals are preserved. These bones are roughly triangular in dorsal aspect and compose part of the skull roof above the orbits. The contact between the supraorbitals and frontoparietals is marked by grooves, which are deeper at their posterior limits. The supraorbitals are also partially covered by the posterior extension of the premaxillae.

Frontoparietal. There is no visible distinction between the frontals and parietals, but a clear suture line divides the left and right elements of these bones. The frontoparietals form almost the entire skull roof above the orbits and the upper temporal fenestrae, being overlaid anteriorly by the slender posterior extension of the premaxillae that projects between the left and right frontoparietals. Above the upper temporal fenestrae, the dorsal margin of the frontoparietals forms a short crest that probably provided a greater area of origin for the musculus adductor mandibulae externus.

Jugal. Only the right jugal is preserved. This is a robust element, mostly composed of three strong processes that contribute to the boundaries of several skull openings. The maxillary process of the jugal extends anteriorly, forming part of the posteroventral margin of the nasoantorbital fenestra as well as it contributes to the lateral margin of the palatal subtemporal fenestra. This process is broken in the preserved jugal of AMNH 22555, preventing an estimation of how far anteriorly the contact with the maxillae was located. The lacrimal process of the jugal is directed dorsally, with a slight anterior inclination, and forms part of the anterior margin of the orbit, as well as part of the posterior margin of the nasoantorbital fenestra. The spot where this process connects with the main corpus of the jugal is depressed, forming a distinct lateral shelf, so that the whole process is medially displaced with respect to the remainder of the bone. The contact with the lacrimal occurs at about one fourth of the total height of the orbit. The most developed jugal process is the posterior, postorbital, one. This bony extension is very thick anteriorly, but becomes narrower throughout its posterodorsal end, where it contacts the postorbital via an overlapping joint. The postorbital process of the jugal composes most of the posterior edge of the orbit, and the whole anterior border of the lower temporal fenestra.

Postorbital. Both postorbitals are preserved; the left one is completely displaced from its original position and the right one shows a slight medial displacement. These bones have a roughly triangular outline and occupy a central position on the temporal region of the skull. The postorbitals make contact dorsally with the frontoparietals, anteriorly with the posterior processes of the jugals and posteriorly with the squamosals. The edges of these bones contribute to the margins of both the upper and lower temporal fenestrae, and also have a small participation in the posterior borders of the orbits.

Lacrimal. In AMNH 22555, only the right lacrimal is preserved. This bone is triangular in shape, making contact with the prefrontal and the nasal dorsally and overlying the lacrimal process of the jugal ventrally. The posterior edge of the lacrimal bears a well-developed, lateromedially broad process directed inside the orbit. The lacrimal is pierced by a vast foramen for the exit of the naso-lacrimal duct, which occupies most of the main corpus of this bone.

Squamosal. The squamosal is a curved bone, with its concavity directed anteriorly, where this element comprises most of the posterior border of the lower temporal fenestra. Dorsally, the squamosal contacts the postorbital and frontoparietals. Between these bones there is a smaller concavity that bounds the ventral margin of the upper temporal fenestra. The squamosal ends ventrally with two acute processes. The anterior one sutures with the slim quadratojugal, whereas the posterior one runs parallel to the quadrate and is probably the origin site of the musculus depressor mandibulae. The posterior, convex edge of the squamosal makes contact with the opisthotic.

Quadratojugal. This slender bone makes contact with the main corpus of the jugal anteriorly and with one of the ventral processes of the squamosal posteriorly, delimiting ventrally the lower temporal fenestra.

Quadrate. Only the right quadrate is completely preserved. This bone contacts the squamosal, quadratojugal and part of the jugal. The anteroventral end of the quadrate expands to form the helical articular surface with the lower jaw. The quadrate shaft runs medially, parallel to the ventral extension of the squamosal. The inclination of the quadrate with respect to the ventral margin of the maxilla is about 145 degrees.

Supraoccipital. The supraoccipital is a broad plate that forms a large portion of the occiput. Above the dorsal margin of the foramen magnum, this bone develops a low sagittal crest, probably linked to the origin of the musculus rectus capitis. Lateral to the crest, the supraoccipital is pierced by two large pneumatic foramina. The dorsal border of the right posttemporal fenestra is preserved, showing that this opening was inclined downwards (Fig. 5).

Opisthotic. In AMNH 22555, both the right and left opisthotics are broken and displaced from their original positions. Although the right element is better preserved, little anatomical information can be drawn from this bone. It can be observed that the opisthotics were configured as wide plates that occupied a considerable portion of the occiput.

Palatine. The structures traditionally regarded, in most pterosaurs, as the palatines were recently reinterpreted as a secondary surface formed by ventral plates of the maxillae (see Ősi et al., 2010; Pinheiro & Schultz, 2012). The high degree of synostosis, common in Pterodactyloidea, makes the individualization of palatal elements difficult. In anhanguerids, the palatines probably bordered the suborbital fenestrae medially, the right element being partially preserved in AMNH 22555 (Fig. 4C; Pinheiro & Schultz, 2012: Figs. 4C and 4D).

Pterygoid. Although most of the posterior palatal bones were lost, part of the left pterygoid lies in dorsal view inside the nasoantorbital fenestra. This bone shows a very long and acute rostral process connected to a concave surface, which is followed posteriorly by a transversal ridge. This ridge can be interpreted as part of a vestigial ectopterygoid, already reported for other specimens of Anhanguera (Pinheiro & Schultz, 2012). In close association, there is a flat, triangular bone of uncertain affinities. It is possible that it represents the posterior extension of the pterygoid, which would contact the basipterygoid caudally.

Vomers. The fused vomers form a slim element that partially divides the choanae medially. There is no sign of sutures between the two vomers or between them and other elements of the palate.

Dentition. Only the dentition pattern of the upper jaw of AMNH 22555 can be assessed, and it is reminiscent of that seen in other species of Anhanguera. The 1st pair of teeth is located at the tip the rostrum, slightly higher than the 2nd pair, facing anteriorly, as is usual in anhanguerians (Rodrigues & Kellner, 2013). The alveoli grow in width until the 3rd pair. As is usual in the genus Anhanguera, the 4th and 7th pairs of alveoli are larger than the 5th and 6th. From the 8th onwards, the alveoli tend to gradually decrease in width. The distances between the alveoli increase gradually, but are most notably larger from between the 7th and 8th alveoli onwards. As noted above, the maxillary margin is not well preserved and most posteriormost alveoli cannot be assessed, but the dentition would continue until at least the beginning of the nasoantorbital fenestra. Some teeth are preserved, showing a curved and pointed shape and longitudinal ridges where the enamel is present, as typical of anhanguerids (Rodrigues & Kellner, 2010).

Discussion

Patterns of premaxillary crest growth in Anhanguera and their taxonomic significance

Morphology of cranial crests has been invariably used as a crucial character in the diagnoses of every single putative species of Anhanguera proposed thus far. Among crest features suggested to distinguish Anhanguera species, the most common is its dorsoventral height and the antero-posterior extension. The first description of Anhanguera blittersdorffi by Campos & Kellner (1985) mentioned a “large sagittal crest on the anterior part of the skull, situated on the premaxillas (sic), which ends almost at the beginning of the external naris” (p. 459). Similarly, Anhanguera spielbergi was described as differing from other species for having a “large premaxillary sagittal crest, in ratio length-total length skull (sic), which extends dorsally from the anterior aspect until the anterior border of the nasoantorbital fenestra” (Veldmeijer, 2003, p. 43). Also, following the taxonomic revision provided by Kellner & Tomida (2000), the only feature that would distinguish “Anhanguera robustus” from other species of this genus would be a large dentary crest with an anterior margin forming an angle of about 50° with the dorsal margin of the lower jaw (Kellner & Tomida, 2000, p. 117).

At least one species assigned to Anhanguera would apparently be diagnosed by a small, rather than a large premaxillary crest: according to Kellner & Tomida (2000), Anhanguera piscator would differ in having a long but low premaxillary crest, which does not reach the highest point of the skull (Kellner & Tomida, 2000, p. 7). According to Kellner & Tomida (2000), the two remaining proposed species of the genus, “Anhanguera araripensis” and “Anhanguera santanae”, would be distinguished by the antero-posterior extension of the premaxillary crest. In “Anhanguera araripensis”, the premaxillary crest would be positioned “right in front of the nasoantorbital fenestra” (p. 105), whereas in “Anhanguera santanae” the premaxillary crest would not reach the anterior margin of the nasoantorbital fenestra, being thus “confined to the anteriormost portion of the skull” (p. 109).

Our regression analysis, however, challenges the use of height and anteroposterior extension of the premaxillary crest as robust characters in the diagnosis of anhanguerids at the species level. As demonstrated here, anhanguerid skulls show statistically significant positive allometric growth of the premaxillary crest (see also the work of Bantim, Saraiva & Sayão, 2015). Besides a simple increase in height, the detected pattern of allometric growth also indicates an anteroposterior development of the premaxillary crest following the increase in total skull size (a pattern also corroborated by the analyses of Bantim, Saraiva & Sayão, 2015).

Following the recent discovery of crested pterosaur assemblages preserving a large number of individuals belonging to a single species (Manzig et al., 2014; Wang et al., 2014), it was determined that pterosaur cranial crest development may indeed be strongly controlled by ontogeny and/or sexual dimorphism, as has been suggested previously (for instance, Bennett, 1992). The strong positive allometric growth of the premaxillary crests in pterosaurs such as Caiuajara dobruskii (Manzig et al., 2014) and the sexual dimorphism related to the premaxillary crest observed in Hamipterus tianshanensis (Wang et al., 2014) are strong evidence to support the idea that pterosaur premaxillary crests evolved through a mode of sexual selection, as has previously been proposed in several studies (e.g., Hone, Naish & Cuthill, 2012; Knell et al., 2013). As is characteristic of sexually selective display structures, it is expected that cranial crest size and morphology were strongly intraspecifically variable in pterosaurs. On these grounds, and in agreement with the results presented here, we propose that premaxillary crest characters should be excluded as diagnostic of pterosaur nominal species without more explicit state delimitation boundaries, and at least when the variation does not imply deep changes on the skull architecture, which is not the case for Anhanguera.

The taxonomy of Anhanguera

On the diagnosis of Anhanguera

Kellner (2003) listed synapomorphies of the genus Anhanguera as (1) the presence of an elongate and medially placed nasal process, (2) a foramen on the nasal process, (3) a characteristic size difference in the rostral teeth (in which the 5th and 6th tooth pairs are smaller than the 4th and 7th ones); (4) scapulae length at most 80% of that of the coracoids, (5) a coracoidal articulation surface with the sternum oval and with a posterior expansion, and (6) a pneumatic foramen on the proximal dorsal surface of the humeri. However, more recently described specimens challenge some of these features and show that they are more widespread among dsungaripteroid pterosaurs. Characters (1) and (2) are present on Ludodactylus sibbicki from the Crato Formation (Frey, Martill & Buchy, 2003), and characters (4), (5) and (6) are also found in Brasileodactylus sp. (SNSB-BSPG 1991 I 27; Veldmeijer, Meijer & Signore, 2009) and in Istiodactylus (Hooley, 1913; Andres & Ji, 2006). Therefore, from these, only character (3) would be unambiguously synapomorphic for Anhanguera.

Naturally, these are characters used in a cladistic sense, but others have also been proposed as diagnostic of the genus. While comparing Anhanguera and Coloborhynchus, Fastnacht (2001) stated that Anhanguera possesses (1) a premaxillary crest beginning more posteriorly instead of at the anterior tip of the rostrum, (2) a premaxillary crest lower than in Coloborhynchus with its height about one third of its length, (3) a thin crest, (4) the anterior end of the rostrum inclined at an angle of about 45 degrees, and (5) the absence of a spoon-shaped distal expansion of the rostrum. From these, our analyses demonstrate that characters (1) and (2) could be attributed to ontogenetic development in the genus Anhanguera. Character (5) is a misinterpretation since the type species, Anhanguera blittersdorffi, has a distal expansion with this morphology (see Rodrigues & Kellner, 2008). Characters (3) and (4), although useful to distinguish Anhanguera from Coloborhynchus, are also present in Liaoningopterus and Caulkicephalus (Wang & Zhou, 2003; Steel et al., 2005; Rodrigues et al., 2015) and therefore are more widespread within anhanguerids. A very similar set of characters was also discussed by Veldmeijer (2003). This author suggested that AMNH 22555 is a juvenile Coloborhynchus. However, some of the characters used by him to separate Anhanguera and Coloborhynchus, such as the position of the premaxillary crest, are also listed as being possibly explained by ontogenetic variation, a view that is supported by our results. Veldmeijer (2003) also suggested that features present at the posterior part of the skull of AMNH 22555 are more similar to Anhanguera spielbergi (regarded by him as belonging to the genus Coloborhynchus) than to the holotype of “Anhanguera santanae”. However, the diagnostic value of these minor differences is dubious. Therefore, it seems that Anhanguera remains diagnosed by a single unambiguous character state, the 5th and 6th tooth pairs being smaller than the 4th and 7th ones, and by combinations of characters.

Here we suggest the following revised diagnosis for Anhanguera, which incorporates the ontogenetic changes discussed above: anhanguerid pterosaurs with premaxillary and dentary median crests at least in advanced ontogenetic stages; premaxillary crest thin; premaxillary crest largely asymmetric; premaxillary crest begins near but not at the tip of the skull; premaxillary crest not confined to the anteriormost tip of the skull; premaxillary crest grows allometrically in height and length during ontogeny; 5th and 6th upper dental alveoli smaller than the 4th and 7th ones; parietal crest blade-like and thin; palatal ridge modest in depth.

AMNH 22555 cannot be confidently referred to what is known as “Anhanguera santanae”

When first described by Wellnhofer (1991), AMNH 22555 was referred to “Anhanguera santanae”, a pterodactyloid pterosaur described a few years before by the same author and from the same formation (Wellnhofer, 1985). The assignment of AMNH 22555 to “A. santanae” (then regarded as “Araripesaurus”; see Introduction) was made mainly on the basis that both specimens share the same number of bones in the carpal series, besides possessing similar sized skulls, even though the position of their premaxillary crests differed (Wellnhofer, 1991). A close examination of the “A. santanae” holotype (SNSB-BSPG 1982 I 90) and comparison to other skulls now known, however, has revealed to us that AMNH 22555 cannot be confidently referred to this species more than to any other proposed species of Anhanguera.

Although AMNH 22555 is indeed similar to the “Anhanguera santanae” holotype in size and overall skull morphology, the two specimens differ in a series of features (Fig. 5). First of all, the frontoparietals of “A. santanae” are relatively narrower and project posterodorsally as a thick frontoparietal crest. On the other hand, the frontoparietals of AMNH 22555 are broader and form a much more delicate crest, which is mostly posteriorly extended. The two specimens also differ in the morphology of the jugal: the lacrimal process of this bone is much broader in A. santanae than in AMNH 22555.

Differences between AMNH 22555 and “A. santanae” also extend to the occipital and palatal regions. In occipital view, it is notable that the supraoccipital crest is much more conspicuous in “A. santanae” than in AMNH 22555. Also, although the occiput of AMNH 22555 is not well preserved, the dorsal margin of the posttemporal fenestra is well marked and reveals that this opening was probably directed downwards, unlike the condition observed in the “A. santanae” holotype. As a consequence of the poor preservation, however, this characteristic must be regarded with caution. In palatal view, it is remarkable that in AMNH 22555 the fusion of the palatal plates of the maxillae develops into a strong palatal ridge (although not as deep as observed in Tropeognathus) that is followed posteriorly by a slight convexity of the palatal occlusal surface. “A. santanae” also bears a palatal ridge, but this structure is much less pronounced and extends posteriorly to a region closer to the choanae than that seen in AMNH 22555. In addition, the choanal morphology is also different between the specimens, with those of AMNH 22555 being distinctly rounder and more lateromedially expanded.

In spite of these remarkable differences between AMNH 22555 and the “Anhanguera santanae” holotype (SNSB-BSPG 1982 I 90), none of the characters listed above have had their distributions well mapped for Anhanguera, and may fall within the range of intraspecific variation for this genus. In addition, it is noteworthy that allegedly diagnostic features of Anhanguera nominal species are, in most cases, subtle and poorly defined, especially those which are related to the presence and morphology of the premaxillary crest. As discussed, the premaxillary crest shows significant allometric growth within Anhanguera-like pterodactyloids, demonstrating that this structure is at least partially body size-dependent and therefore has limited use for taxonomic purposes. Bearing this in mind, we reassess here the significance of anatomical features of the premaxillary crest traditionally thought to support Anhanguera species, and elucidate the impact of this on the taxonomy of this genus.

On the validity of “Anhanguera santanae” and other species of Anhanguera

A reappraisal of the purportedly diagnostic features of the individual Anhanguera species revealed that most, if not all, of the characters that are currently used to define the separate species are probably well inside the range of intraspecific variation. Considering this, it is pertinent to inquire about the validity of each one of the species attributed to this genus.

When first described, “Anhanguera santanae” was differentiated from other pterosaurs on the basis of characters that are today known to be widely distributed among other Santana Group ornithocheiroids. A complete discussion of the validity of the diagnostic features originally proposed for “A. santanae” was made by Kellner & Tomida (2000). As a conclusion, these authors stated that the only remaining diagnostic character for this taxon would be the position of the premaxillary crest, well anterior to the nasoantorbital fenestrae. It is noteworthy that the premaxillary crest itself is not preserved on “A. santanae” holotype, and its presence is inferred by the acute dorsal margin of the premaxillae close to the anterior extremity of the specimen, as preserved. One of the specimens analyzed in the present allometric regression, SMNK PAL 1136, presents a premaxillary crest that can be presumed to be positioned as far from the nasoantorbital opening as inferred in the holotype of “A. santanae”. As discussed, premaxillary crest characters are here regarded as unfit for the diagnosis of nominal anhanguerid species, what means that “A. santanae” holotype lacks unambiguous diagnostic features and should be considered as a nomen dubium.

Anhanguera blittersdorffi, the type species of Anhanguera, was first diagnosed by characters that later proved to be diagnostic of more inclusive clades, such as the presence and morphology of the premaxillary and frontoparietal crests and the presence of a distal expansion and of larger teeth at the tip of the rostrum (Campos & Kellner, 1985). Actually, A. blittersdorffi has the standard morphology of Anhanguera and, after the description of other anhanguerids, it became difficult to recognize unique characters for this species. A subsequent revision (Kellner & Tomida, 2000) proposed diagnostic characters of the species as a “lower skull with a proportionally shorter quadrate”. Those characters, however, are subjective and ambiguous, and lack a quantified definition to delimit the state boundaries. Besides the holotype, only one additional specimen has been formally attributed to A. blittersdorffi (Pz-DBAV-UERJ 40) (Kellner & Tomida, 2000), though the latter still lacks an anatomical description. Remarkably, the number of alveoli on A. blittersdorffi upper jaws (52) is higher than in any other proposed Anhanguera species and this might be a more suitable diagnostic character for this taxon.

Figure 6 Overview of the holotypes of several Anhanguera species.

(A) Anhanguera blittersdorffi (MN 4805-V) in lateral view. (B, C, F), “Anhanguera araripensis” (SNSB-BSPG 1982 I 89) in dorsal, ventral, and lateral views, respectively. (D) detail of (C); arrow points a lateral projection of the pterygoid. (E) detail of Tropeognathus mesembrinus holotype (SNSB-BSPG 1987 I 46); arrow points a bulge laterally on the pterygoid. (G, H) holotype of “Anhanguera robustus” (SNSB-BSPG 1987 I 47) in dorsal and lateral views, respectively. (I) holotype of Anhanguera spielbergi (RGM 401 880) in lateral view.

“Anhanguera araripensis” was described based on a very incomplete skull with associated postcranial bones by Wellnhofer (1985). As was the case for A. blittersdorfii, “A. araripensis” was first diagnosed by characters that later were demonstrated to be more widespread among anhanguerids or dependent on ontogenic status of specimens. After the revision of Kellner & Tomida (2000), only two characters remained as diagnostic for this species: the dorsal margin of the premaxillae being “keel shaped” up to the anterior end of the nasoantorbital fenestrae (a character described as being related to the position of the premaxillary crest, which is not preserved at the holotype), and the presence of small lateral projections on the basioccipital processes of the pterygoids (Kellner, 1991) (Fig. 6D). Based on this character, other specimens have also been referred to this species, such as MN 4735-V (Kellner & Tomida, 2000) and SAO 16494 (Veldmeijer, 2003; Veldmeijer, 2006). We agree that the “keel shaped” dorsal margin of the premaxillae is probably related to the presence and morphology of the premaxillary crest and, for the reasons described above, challenge the taxonomic value of this character. Regarding the lateral projections of the pterygoids inside the subtemporal fenestrae, we consider this character as problematic, because it is probably related to the bone growth between different elements of the adductor musculature that crossed the subtemporal openings. Also, these projections are exceptionally delicate and were probably abraded on not so well preserved skulls. Remarkably, specimens such as the holotype of Tropeognathus mesembrinus (SNSB-BSPG 1987 I 46) and A. blittersdorffi (MN 4805-V) have very discrete bulges at this same location (Fig. 6E). Thus, we here regard the holotype of “A. araripensis” as nondiagnostic and, for this reason, “Anhanguera araripensis” should also be considered as a nomen dubium.

“Anhanguera robustus” was originally referred to the genus Tropeognathus by Wellnhofer (1987) and later assigned to Anhanguera (Kellner & Campos, 1988). This taxon was originally diagnosed by the presence of a well-developed dentary crest, with a straight anterior margin; and by a spoon-like anterior expansion of the dentaries and long anterior teeth. As has already been observed by Kellner & Tomida (2000), strong anterior teeth associated to a lateral expansion of the dentaries are considered to be widespread among anhanguerids. The other supposedly diagnostic characters are related to the dentary sagittal crest and are probably associated to the apparently advanced ontogenetic stage of the specimen (SNSB-BSPG 1987 I 47). Thus, we also consider “Anhanguera robustus” as a nomen dubium.

Despite its large body size, the holotype of Anhanguera piscator presents clear evidence of an early ontogenetic stage, which partially explains the presence of the low premaxillary crest that was regarded by Kellner & Tomida (2000) as diagnostic for this species. Our analysis demonstrates that premaxillary crest height in this species cannot be explained by allometric growth alone, but nonetheless we consider this character alone to be inappropriate for the diagnosis of anhanguerids. Kellner & Tomida (2000) indicated another cranial character as diagnostic for this taxon: a “basisphenoid constricted in the middle part” (Kellner & Tomida, 2000, p. 7). This feature cannot be accessed in SNSB-BSPG 1982 I 89 (“Anhanguera araripensis”) or SNSB-BSPG 1987 I 47 (“Anhanguera robustus”). Although this character still lacks an unambiguous morphometric definition, basisphenoid morphology in A. piscator holotype is indeed different from what is observed in Anhanguera blittersdorffi and SNSB-BSPG 1982 I 90 (“Anhanguera santanae”), resembling the condition of Anhanguera spielbergi. Other proposed diagnostic features of A. piscator are associated to the postcranial skeleton, which is poorly preserved or absent in most other Anhanguera holotypes. A. piscator is here retained as a valid taxon, at least until more information about the distribution of these postcranial characters and the basisphenoid morphology becomes clearer within Anhangueridae.

Veldmeijer (2003) considered Anhanguera spielbergi to be a representative of Coloborhynchus, including in the diagnosis of this species an “ill-defined, almost absent (…) palatinal ridge and corresponding mandibular groove; mandibular groove not extending onto spoon-shaped expansion; slight, almost absent, ventrolaterally extending tooth-bearing maxillae; large premaxillary sagittal crest, in ratio length-total length skull, which extends dorsally from the anterior aspect until the anterior border of the nasoantorbital fenestra; strongly medial bended rami; sternum with rounded triangular posterior plate of which the length is as long as the width” (Veldmeijer, 2003, p. 43). Although the palatal ridge of A. spielbergi is indeed weaker than that which is observed in other Anhanguera holotypes, it is still not clear how this character is affected by ontogeny, the same also being a potential issue for the mandibular groove morphology. As we have discussed, premaxillary crest morphology is here regarded as inappropriate for species-level diagnoses. Furthermore, a medial bending of mandibular rami cannot be assessed in most of the other holotypes, but is present in other complete anhanguerid mandibles (for instance, “Anhanguera robustus”). Although the intrageneric variation of the remaining characters is still unclear, we regard A. spielbergi as a valid taxon, a taxonomic statement that requires further testing through more comprehensive sampling within the genus.

A highly diverse genus or an exceptionally biased record?

Specimens attributable to Anhanguera often present slight differences on their skull anatomies, especially with respect to the size and morphology of the premaxillary crest. Historically, these different morphotypes were used to base the definition of new taxa, which at the time was not necessarily incorrect, given the fact that our knowledge about ontogenetic and sexual variability connected to crest morphology was (and still is) incipient. Nowadays, however, this practice has resulted in an abundance of nominal species with, as we demonstrate, continuous morphologies. As a result, it is considerably difficult to attribute any new material to a previously described species with any proper degree of certainty. This same issue was detected before in other fossil localities that have, historically, yielded pterosaur fossils, such as the Niobrara and Pierre Shale formations of the USA (Pteranodon and Nyctosaurus sites) and the Solnhofen limestones of Bavaria, Germany. Similarly to what we discuss here for Anhanguera, the diversity of taxa found in those sites has been reassessed taking into account the influence that ontogeny, sexual dimorphism, individual differences and time may have on morphological disparities that have previously been considered to be of taxonomic significance (e.g., Bennett, 1992; Bennett, 1994; Bennett, 1995).

A possible overestimation in the anhanguerid diversity of the Romualdo Formation was also already pointed out by Kellner & Tomida (2000). These authors commented on the lack of comparable elements between some of the taxa and on potential intraspecific variations for the taxonomic inflation, although not making reference to other potential biases.

As the relation between morphological disparity and speciation is vague, the application of the prevailing definition of the biological species concept (grounded on reproductive isolation) to the fossil record is exceedingly challenging (e.g., Gingerich, 1985; Bennett, 1994; Kellner, 2010). This is even more delicate when one is dealing with lineages that lack extant analogues or direct descendants, as is the case for pterosaurs. In order to distinguish fossil and extant species, the amount of morphological variation among studied specimens is less important than the presence of morphological discontinuities (Gingerich, 1985). Disparate morphologies that show continuous intermediates in the sample are, thus, better explained by intraspecific variation or temporal evolutionary effects (this later only recognizable in the fossil record).

As we demonstrated, most of the allegedly diagnostic characters traditionally used to distinguish proposed Anhanguera species display continuous variation in the available sample pool and are correlated to skull size, and as a result are generally unfit for taxonomic purposes. The detected residual variation (not attributable to the allometric growth of the skull) is, in most of the cases, characterized by disparate conditions linked by intermediate morphologies. However, in some other cases, as the premaxillary crest morphology of specimen MN 4735-V and Anhanguera piscator holotype illustrates, the residual variation is well beyond the condition expected for animals of their sizes, and therefore perhaps more reflective of taxonomic discrepancies. We discuss, here, possible explanations for this peculiar pattern of morphological disparity of Anhanguera-like pterosaurs.

A natural ecological question that follows the assumption that Romualdo Formation pterosaur taxa were sympatric and coeval, is how such a large number of taxa with supposedly overlapping ecological niches may have coexisted. However, competitive exclusion of species happens only when the resources are scarce to the point of limiting population growth. If we assume, as is likely, that Anhanguera species competed for prey, sufficiently high fish populations could sustain several sympatric piscivorous species. This, however, would result in an apparently aberrant community structure, and the pattern observed in the fossil record may be better explained by the influence of biological and stratigraphic bias.

Although our allometric regressions are not per se direct evidence that premaxillary crests grew with age, the strong correlation of crest development with respect to skull size makes it very likely that the patterns observed here indeed reflect an ontogenic growth trajectory. Allometric growth of skull ornaments in pterosaurs was recently confirmed by the discovery of monospecific bonebeds with fairly complete growth series (e.g., Manzig et al., 2014). The strong positive allometry demonstrated here (as in pterosaurs like Caiuajara dobruskii) is characteristic of sexually selected traits (Tomkins et al., 2010), which are exceptionally variable within species. Thus, it is likely that a considerable amount of the morphological disparity observed in anhanguerids is attributable to intraspecific variation. Sexually selected characters tend also to be sexually dimorphic, and sexual dimorphism related to cranial premaxillary crests was present in pterosaurs (e.g., Wang et al., 2014). It is possible that anhanguerid premaxillary crests were also sexually dimorphic, which would explain at least some of the residual variation recovered by our analyses. However, small sample size and the probable effect of stratigraphic biases (as we discuss below) makes it impossible to assess this hypothesis at the time. It is consensual that robust synecological inferences based on Romualdo Formation fossils are impossible based on museum specimens alone. The reason for this hindrance is that the commercial exploitation of Romualdo Formation fossil bearing strata unfortunately disregards important field data, such as those concerning the stratigraphic distribution and abundance of species. Virtually all the Romualdo Formation specimens deposited in museums and universities throughout the world (i.e., those available for scientific research) fall under this scenario. The high commercial value of complete specimens or specific taxa, such as pterosaurs, created a strong collection bias and, as a result, museum specimens are not representative of the actual Romualdo Formation diversity (Fara et al., 2005; Vila Nova et al., 2011). Stratigraphically controlled excavations on Romualdo Formation are still incipient (Fara et al., 2005; Vila Nova et al., 2011). The few works dealing with the results of these enterprises, however, have already demonstrated the presence of strong geographic and stratigraphic biases, which may impact upon our understanding of Romualdo Formation pterosaur taxonomy and diversity.

The yet incipient results derived from controlled excavations on the Romualdo Formation already demonstrate clear evidence for faunal turnover, through the substitution of a basal fish assemblage dominated by the gonorynchiform Tharrhias by upper strata where the most abundant taxon is the aspidorhynchid Vinctifer (Fara et al., 2005). Possible reasons for this faunal interchange have still not been investigated. However, considering the presumably low deposition rate of the shales that embed Romualdo Formation fossil concretions, it is likely that a substantial time interval was associated with this turnover.

The temporal resolution of Romualdo Formation fossils was never estimated and several events of mass mortality probably took place (Fara et al., 2005; Vila Nova et al., 2011). Thus, based on the present state of knowledge, it is likely that at least some of the Romualdo Formation pterosaurs were not coeval. This could also be an explanation for the apparently high number of similar species of anhanguerians in the same geological unit, since we might have a sample that includes species separated in time. Thus, it is possible that different Anhanguera-like morphotypes may represent subtle morphological changes in a lineage undergoing anagenetic evolution. A similar pattern was proposed by Bennett (1994) for different Pteranodon species (but see Kellner, 2010). Stratigraphically controlled excavations, such as the ones reported by Fara et al. (2005) and Vila Nova et al. (2011) hopefully will shed light on this issue.

Conclusions

Even though more than a dozen relatively complete skulls referable to the Anhangueridae and closely related taxa are nowadays held in public collections, this is the first study to perform a comprehensive morphometric analysis of continuous morphological features seen in the skulls of members of this clade. As a result, characters related to both dorsoventral height and the anteroposterior extension of the premaxillary crest are found to be allometrically correlated to skull size, and therefore at least in part to ontogeny. The observation that anhanguerid premaxillary crest morphology is size-dependent also means that it is largely unfit to be used as a diagnostic character for delimiting species, as has been commonly proposed for this group in the past. A taxonomic review excluding these characters reveals that as few as three Anhanguera species are potentially valid: A. blittersdorffi, A. piscator and A. spielbergi. The significance of the minor, continuous differences between specimens is still not entirely clear, though. Controlled stratigraphic studies on the Romualdo Formation demonstrate evidence of faunal turnover in fishes, and the same could be true also for pterosaurs. Thus, the seemly continuous morphological changes observed in anhanguerids could possibly be explained by anagenetic evolution. However, as virtually all pterosaur specimens from this unit lack fundamental stratigraphic information, it is impossible to test this hypothesis at the present.

Supplemental Information

Data S1 Raw data for morphometric analyses

TPS file containing MorphoJ-friendly raw data for the performed morphometric analyses.

Click here for additional data file.

For granting access to AMNH 22555 and other pterosaur specimens, the authors are particularly indebted to Mark Norell and Carl Mehling (AMNH). Also, we would like to thank the following people for allowing the study of specimens under their care and for kind help during visits: Álamo Saraiva and João Kerensky (MPSC); Oliver Rauhut and Markus Moser (SNSB-BSPG); Alexander Kellner and Helder Silva (MN); Eberhard Frey (SMNK); Rainer Schoch (SMNS); Sandra Chapman and Lorna Steel (NHMUK); Dan Pemberton and Matt Riley (CAMSM); Stephen Hutt (IWCMS); David Gelsthorpe (MANCH); Urs Oberli (SAO); Jon de Vos (Naturalis); Mauro Bon (MSN); and Wang Xiaolin (IVPP). FLP thanks Cesar Schultz for advisement on pterosaurs and Jose R.I. Ribeiro for initial advising in morphometric methodology, TR thanks Alexander Kellner for advisement and support. We also acknowledge Pedro Godoy (University of Birmingham) for comments and suggestions on the methodology applied here, Jonathan Tennant for suggestions and review of the language, as well as Renan Bantim and an anonymous referee for valuable comments on an early version of this paper.

Institutional abbreviations

AMNH American Museum of Natural History, New York, USA

MHNS Museu de História Natural de Sintra, Sintra, Portugal

MN Museu Nacional/Universidade Federal do Rio de Janeiro, Rio de Janeiro, Brazil

MPSC Museu de Paleontologia, Santana do Cariri, Brazil

NHMUK Natural History Museum, London, UK

NSM National Science Museum, Tokyo, Japan

DBAV-UERJ Universidade do Estado do Rio de Janeiro, Rio de Janeiro, Brazil

RGM National Natuurhistorisch Museum/Naturalis, Leiden, The Netherlands

SAO Sammlung Oberli, a private collection belonging to Mr. Urs Oberli, Sankt Gallen, Switzerland

SMNK Staatliches Museum für Naturkunde, Karlsruhe, Germany

SNSB-BSPG Staatliche Naturwissenschaftliche Sammlungen Bayerns/Bayerische Staatssammlung für Paläontologie und Geologie, Munich, Germany

Additional Information and Declarations

Competing Interests

Author Contributions

Data Availability

The authors declare there are no competing interests.

Felipe L. Pinheiro conceived and designed the experiments, performed the experiments, analyzed the data, wrote the paper, prepared figures and/or tables, reviewed drafts of the paper.

Taissa Rodrigues conceived and designed the experiments, analyzed the data, wrote the paper, reviewed drafts of the paper.

The following information was supplied regarding data availability:

The raw data is included as a Supplemental File.

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
