# Peer review of "Anhanguera taxonomy revisited: is our understanding of Santana Group pterosaur diversity biased by poor biological and stratigraphic control?"

_PeerJ, doi:10.7717/peerj.3285_

## Round 0.1 · original submission · Minor Revisions

· Academic Editor

Minor Revisions

Dear authors,

I am pleased to inform you that we have now two review reports about your manuscript “Anhanguera taxonomy revisited: is our understanding of Santana Group pterosaur diversity biased by poor biological and stratigraphic control?”. Both reviewers considered that your article should be published in PeerJ after minor revisions which need to be addressed. Thus please, consider to revise all the recommendations and concerns from the reviewers and submit a new version of your manuscript. Otherwise, provide a rebuttal letter explaining your points in the case you disregard them.

Please, note that the reviewers also submitted annotated pdf files with suggestions that also need to be taken into account.

My recommendation is that you insert labeling into some of the figures to identification of the main structures and bones displayed. Maybe your Fig. 5 can be useful for that purpose.

I hope you will find the reviews interesting for the improvement of your manuscript, looking forward to see the updated version very soon.

Best wishes,
Graciela Piñeiro

Reviewer 1 ·

Basic reporting

Some very minor problems with the English language and workding - see comments below.

Experimental design

This is an interesting paper and I have not detected any methodological problem.

Validity of the findings

This contribution addresses a complex question regarding the several species of pterosaur that were named from the Romualdo FM and is an interesting ms. Please see some of the comments below.

Additional comments

The present ms is a very interesting paper that contributes to the discussion about the pterosaur diversity of one of the most interesting deposits worldwide - the Romualdo Formation, that is part of the Santana Group. To make a long story short, I am not sure if I do agree with all points made by the authors, but I welcome the opportunity that they make in shedding some new light on the - lets say - "inflated" pterosaur fauna from this deposit. By the way, there is no question that a more detailed description of the AMNH A. santanae was necessary and long doe, and they do this competently.

The authors presented also do present some new and interesting arguments (e.g., the questions about these species living in distinct time periods that might not be recognized due to lack of collecting information) that will have to be addressed by others that might not agree with their conclusions, but the way they did that was very professional and well-founded. There are - as usual - some points that might be raised and that the authors should/might like to discuss. I also do provide some suggestions (and questions that I think could be addressed) and invite them to use them as they see fit. All a listed below.

To sum up, I think that this is an interesting contribution and I do support its publication although I do not agree with some of their conclusions (agree to disagree).

1) Like always, for non-native speaker there are parts of the text that might need some clarifications and sound somewhat awkward. What do they mean by "abnormal difficulty in referring well-preserved skulls into known taxa"? This is the real world for all areas and difficulties in referring specimens to pre-existing taxa is not uncommon (try to work with frogs or squamates!). I would tune down this point.
Also - they are quite harsh regarding the illegal collected specimens and the loosing of data. Yes, this is true, but see what happens with pterosaurs around the world - in the past and know (e.g., Pteranodon complex, China, Cambrige Greensand). Yes, one has to point out the problem with commercial collecting, but do they think that worlds like "irresponsible collection" (perhaps they want to say "collecting") really do apply for fossils collected some decades ago most by local farmers that lack any training in paleontology? Again, I recommend them to tune down this kind of statements that add nothing for the scientific information they want to discuss.

2) What do they mean by "biological or stratigraphically biased"? Please make clear what you want to state.

3) Yes, there are papers out there on monospecific pterosaur bone beds that have shown that crests varies during ontogeny and in at least one case might be sexually dimorphic. However, I would be interested to hear from the authors if they see any direct evidence of this in the sample they have studied or if they have raised those arguments (in several parts of their ms) only based that those possibilities are theoretically also possible for the sample they have studied.

4) Why the question mark regarding the Albian age for the Romualdo unit? Do they have any reference for that?

5) The palatine has been redefined regarding its position in several specimens and apparently the authors agree to that. However, there are papers out there that contradict or at least show different interpretations regarding the study presented by Ösi et al 2010 and Pinheiro & Schultz 2012 which might at least be mentioned here. In any case, please indicate the location of the palatine in AMNH specimen you describe.

6) What evidence do you have that the ectopterygoid is vestigial? Please indicate this bone in the specimen you describe.

7) What evidences does your material or your study provide that crests evolved due to sexual selection? Perhaps you should cite also the studies that argue against this idea.

8) What do you mean by stating that the anhanguerid skull is very conservative? And if you can argue that (what I am not sure et al), would not the small differences be even more significant if the skull in those animals would not be as conservative as you think? If the latter would be true, why could those features not be diagnostic? Although I tend to agree with the authors in the general points that they have maid, they should develop this a little bit further to be more clear what they mean.

9) I got a little bit confused: so, what about Maaradactylus? Do you think it is valid or not?

10) I know that it will be quite hard to address this, but the authors have made several comments about taxa from the Romualdo Lagerstätte that I think they should also comment on Unwindia. Is this fragmentary rostrum a valid taxon referable to Anhangueridae (as some pointed out) or not?

·

Basic reporting

This article is written in professional English using clear and unambiguous text and is in conform to professional standards of courtesy and expression.

This article include sufficient introduction and background and relevant prior literature appropriately referenced.

The structure of the article is conform to an acceptable format of ‘standard sections’. The figures is be relevant to the content of the article, of sufficient resolution, and appropriately described and labeled, with some exceptions marked in the pdf. I miss the available of raw data of geometric morphometrics.

The submission is ‘self-contained,’ and represent an appropriate ‘unit of publication’, including all relevant results.

Experimental design

This paper carry a original primary research within aims and scope of the PeerJ. This research question well defined, relevant and meaningful and a rigorous investigation performed to a high technical and ethical standard. The methods is be described with sufficient information to be reproducible by another investigator.

Validity of the findings

This research no repite the ‘pointless’ of well-known, widely accepted results.

The data is robust, statistically sound, and controlled, but the data on which the conclusions are based is no provided or made available in an acceptable discipline-specific repository.

The conclusions is appropriately stated and connected to the original question investigated.

Additional comments

This paper brings a new view on the taxonomy of Anhanguera pterosaurs and should be considered for publication with minor revisions.

---

## Round 0.2 · Minor Revisions

· Academic Editor

Minor Revisions

Dear authors,

I have reviewed carefully the last uploaded version of your manuscript and found some issues that need to be solved. I have detailed them in the annotated PDF, attached to this letter. The main concern that I have is about the number of individuals of each species that you revised for the morphometric analysis. I suggest that you add a Table with the information that I described in the annotated PDF. As you will see, this request is important to assess one of your main conclusions about the value of the cranial crests characters in the identification of the species. Other concern is respecting to the no inclusion of A. santanae into the morphometric analysis; maybe it could be good that you include an explanation for the exclusion.

The manuscript is well written to me, I found some minor corrections that I have marked in the annotated PDF, but always is good if you can find an English native speaker that kindly can offer a help for improving the language.

I hope that you consider my recommendations and can upload a revised version of the manuscript soon. You can also refute them providing a rebuttal letter to explain your point of view.

With my best regards,
Graciela Piñeiro

---

## Round 0.3 · Minor Revisions

· Academic Editor

Minor Revisions

Dear authors,
I have kindly seen that your manuscript has been highly improved. I just marked some minor issues that remain unclear to me or poorly explained or need additional information. I have included them in the attached annotated PDF file.

Hope you can solve quickly these requests and submit the new version of your manuscript soon.

Best regards,
Graciela Piñeiro

---

## Round 0.4 · accepted · Accept

· Academic Editor

Accept

Dear authors,

The manuscript has been very much improved, and clearly reflects the high uncertainty that surrounds the taxonomic study of pterosaurs from the Romualdo Formation. I think that this article will contribute as a start point to take into account for future research on taxonomy of new pterosaur records. Although I would like to reiterate the contradictory sense that I see around to “propose that premaxillary crest characters should be excluded as diagnostic of pterosaur nominal species” but defining the genus Anhanguera by the combination of 8 characters, 6 of which are related to the shape and extension of the premaxillary crest, I have decided to accept the manuscript in its current version. My decision was based on the understanding that despite warning about the inconvenience of applying ontogenetically variable characters for taxon identification, at the same time your contribution justifies in certain grade the use of premaxillary crest shape and size by previous authors, because although ambiguous, it seems to be at first hand, an indispensable tool for pterosaur taxonomic identification.

I am glad to have handled this nice and valuable innovative paper. Congrats!

Best regards,
Graciela Piñeiro